# Total Synthesis of Eliglustat via Diastereoselective Amination of Chiral *para*-Methoxycinnamyl Benzyl Ether

**DOI:** 10.3390/molecules27082603

**Published:** 2022-04-18

**Authors:** Younggyu Kong, Pulla Reddy Boggu, Gi Min Park, Yeon Su Kim, Seong Hwan An, In Su Kim, Young Hoon Jung

**Affiliations:** School of Pharmacy, Sungkyunkwan University, Suwon 16419, Korea; kongyounggyu@gmail.com (Y.K.); prboggu@skku.edu (P.R.B.); kimin1216@nate.com (G.M.P.); ols37@naver.com (Y.S.K.); anju4733@naver.com (S.H.A.); insukim@skku.edu (I.S.K.)

**Keywords:** amination, eliglustat, chlorosulfonyl isocyanate, Sharpless asymmetric dihydroxylation, total synthesis

## Abstract

Eliglustat (Cerdelga^®^, Genzyme Corp. Cambridge, MA, USA) is an approved drug for a non-neurological type of Gaucher disease. Herein, we describe the total synthesis of eliglustat **1** starting from readily available 1,4-benzodioxan-6-carbaldehyde via Sharpless asymmetric dihydroxylation and diastereoselective amination of chiral *para*-methoxycinnamyl benzyl ethers using chlorosulfonyl isocyanate as the key steps. Notably, the reaction between *syn*-1,2-dibenzyl ether **6** and chlorosulfonyl isocyanate in the mixture of toluene and hexane (10:1) afforded *syn*-1,2-amino alcohol **5** at a 62% yield with a diastereoselectivity > 20:1. This observation can be explained by competition between the S_N_i and the S_N_1 mechanisms, leading to the retention of stereochemistry.

## 1. Introduction

The 1,2-amino alcohol motif has received considerable attention because of its essential role as a fundamental building block [1,2,3] and its presence in several pharmaceutical agents [4,5,6,7,8,9,10,11,12,13,14,15], including more than 80 FDA-approved drugs [7]. For example, eliglustat (**1**, Cerdelga^®^, Genzyme Corp. United States) has been used for treating Gaucher disease by inhibiting glucosylceramide synthase [4,5,6]. Florfenicol (**2**) is mainly used in veterinary medicine for treating bovine respiratory disease [8,9,10]. Droxidopa (**3**) has been used as a prodrug of neurotransmitter norepinephrine [11,12,13]. Zetomipzomib (**4**, KZR-616), a first-in-class inhibitor of the immunoproteasome, selectively targets the LMP7 and LMP2 subunits of the immunoproteasome [14,15] (Figure 1).

In particular, eliglustat is a medication used to treat a non-neurological type of Gaucher disease and was approved by the United States FDA in 2014 [4,5,6,7]. Due to its potent biological activity and unique structural features, several synthetic approaches for the preparation of eliglustat have been developed [16,17,18,19,20,21]. For example, the Genzyme company reported the synthesis of eliglustat from 1,4-benzodioxan-6-carbaldehyde through a chiral aldol reaction with the chiral-pooled intermediate of phenylglycinol [19]. Xu and coworkers described the total synthesis of eliglustat through a Crimmins aldol reaction of 1,4-benzodioxan-6-carbaldehyde used with Evans auxiliary [20]. Moreover, the synthesis of eliglustat using an organocatalytic asymmetric Henry reaction as a key step was also reported [21]. As part of an ongoing research program aimed at the total synthesis of pharmacologically active compounds via stereoselective amination of chiral benzylic ethers using chlorosulfonyl isocyanate (CSI) [22,23,24,25,26], we herein report the total synthesis of eliglustat from commercially available 1,4-benzodioxan-6-carboxaldehyde via diastereoselective amination of chiral *para*-methoxycinnamyl benzyl ethers using CSI.

## 2. Results and Discussion

The retrosynthetic analysis of eliglustat (**1**) is shown in Figure 1. It shows that **1** was synthesized through ozonolysis, reductive amination, the removal of protected groups, and acylation from the intermediate **5**. The required *syn*-1,2-amino alcohol motif of **5** was prepared via the diastereoselective installation of a NHCbz moiety into *syn*-1,2-dibenzyl ether **6** using CSI, which in turn was easily derived from 2,3-dihydrobenzo[*b*][1,4]dioxine-6-carbaldehyde (7) as a starting material.

Our initial study focused on the efficient construction of *syn*-1,2-dibenzyl ether **6**, which was subjected to a stereoselective amination methodology to give the protected *syn*-1,2-amino benzyl ether **5** (Figure 2). The Horner–Wadsworth–Emmons olefination of **7** with trimethyl phosphonoacetate in the presence of sodium hydride afforded olefin **8** at a 98% yield. The Sharpless asymmetric dihydroxylation [27] of olefin 8 with AD-mix-β furnished the chiral *syn*-1,2-diol **9** at an 82% yield. Next, the *syn*-1,2-diol **9** was protected with acetone to produce **10** at an 83% yield. The reduction of an ester group on **10** to a primary alcohol followed by Swern oxidation and Wittig olefination provided the *para*-methoxycinnamyl derivative **11** (*E:Z* ratio of 1:3) at a 67% yield over three steps. The removal of acetal and the subsequent benzylation of *syn*-diol **12** furnished the *syn*-1,2-dibenzyl ether **6** at a 71% yield.

Next, we screened the optimal reaction conditions for the stereoselective amination of *syn*-1,2-dibenzyl ether **6** using CSI (Table 1). The coupling of **6** and CSI in dichloromethane at 0 °C furnished the corresponding carbamate **5** at a 29% yield with a diastereoselectivity of 1.2:1 (Table 1, entry 1). An examination of the effect of solvents under various temperatures indicated that the improved yield and diastereoselectivity were achieved by using toluene, affording **5** (Table 1, entry 9). After further screening, we found that the highest diastereoselectivity (dr > 20:1) was obtained in a mixture of toluene and *n*-hexane (10:1) at −78 °C with an improved yield of 62% (Table 1, entry 11).

The diastereomeric ratio was determined by ^1^H NMR analysis. As shown in Figure 2 (case selected), a single proton at the chiral position was significantly distinguishable in the ^1^H NMR spectra. As a result, higher diastereoselectivity was obtained in the mixed solvent system of toluene and n-hexane (entry 11) than in other single nonpolar solvent systems with CH_2_Cl_2_ (entry 1 and 2), n-hexane (entry 5) and toluene (entry 9).

The origin of the diastereoselectivity can be explained by competition between the S_N_*i* pathway leading to a retention of stereochemistry through a four-membered transition state and the S_N_1 pathway through a carbocation intermediate [28] (Figure 3). The improved diastereoselectivity in mixed nonpolar solvents (toluene and *n*-hexane) at low temperatures (−78 °C) might be realized by the increased formation of a tight ion pair intermediate (IIA) rather than a carbocation intermediate (IIB).

To complete the total synthesis of **1**, an ozonolysis reaction of compound **5** was performed to afford the corresponding aldehyde, which was converted to the tertiary amine 13 by reductive amination with pyrrolidine (Figure 4). The Pd-catalyzed hydrogenolysis of 13 afforded *syn*-1,2-amino alcohol 14 at an 81% yield. Finally, the acylation of amino alcohol 14 with *n*-octanoyl chloride furnished eliglustat (1) at a 53% yield. The analytical data (^1^H NMR) and melting point of the synthesized eliglustat (1) were in full agreement with those reported in the literature [19].

## 3. Materials and Methods

### 3.1. General Information

Commercially available reagents were used without additional purification unless otherwise stated. All reactions were performed under an inert atmosphere of nitrogen. The nuclear magnetic resonance spectra (^1^H and ^13^C NMR) were recorded on an Agilent 400 MR 400 MHz Spectrometer with Oxford NMR AS400 Magnet spectrometers in CDCl_3_ solution, and the chemical shifts were reported as parts per million (ppm). Resonance patterns were reported with the notations s (singlet), d (doublet), t (triplet), q (quartet), and m (multiplet). In addition, the notation of br was used to indicate a broad signal. The coupling constants (*J*) were reported in hertz (Hz). The IR spectra were recorded on a JASCO FT/IR-4600 spectrophotometer and were reported as cm^−1^. Optical rotations were measured with a JASCO P-2000 polarimeter in the solvent specified. Thin layer chromatography (TLC) was carried out using plates coated with Kieselgel 60 F254 (Merck). For flash column chromatography, *E.* Merck Kieselgel 60 (230–400 mesh) was used. The liquid chromatography-mass spectra (LC/MS) were recorded on an AGILENT 1260 infinity II/InfinityLab LC/MSD system. The high-resolution mass spectra (HRMS) were recorded on a JEOL JMS-700 mass spectrometer. Copies of 1H and 13C NMR Spectra are in the Appendix A.

### 3.2. Experimental Procedure

#### 3.2.1. Synthesis of (*E*)-Methyl 3-(2,3-Dihydrobenzo[*b*][1,4]dioxin-6-yl)acrylate (**8**)

To a solution of sodium hydride (0.67 g, 16.75 mmol) in dry tetrahydrofuran (25 mL), trimethylphosphonoacetate (4.93 mL, 30.46 mmol) was added at 0 °C. The resultant mixture was stirred for 10 min, and a solution of 1,4-benzodioxan-6-carboxaldehyde **7** (2.50 g, 15.23 mmol) in tetrahydrofuran (25 mL) was added. The reaction mixture was stirred at room temperature for 2 h. The reaction mixture was slowly quenched with ice water and extracted with EtOAc (2 × 100 mL). The combined organic layers were washed with brine, dried over anhydrous sodium sulfate, filtered, and concentrated in vacuo. The residue was purified by flash chromatography on silica gel (5~40% EtOAc/*n*-hexanes) to afford **8** (3.27 g) at a 97.5% yield as a white solid (R_f_ = 0.46 (20% EtOAc/*n*-hexanes); mp 66–68 °C (lit, mp 66–68 °C) [29,30]; IR(neat) ν 3905, 3855, 3809, 3730, 3680, 3626, 3556, 3408, 3298, 2947, 2829, 2360, 1714, 1635, 1610, 1579, 1508, 1435, 1309, 1286, 1252, 1174, 1124, 1063, 1032, 916, 887, 856, 814, 650, 492, 463, 432, 422, and 405 cm^−1^; ^1^H NMR (400 MHz, CDCl_3_) [30] δ 7.57 (d, *J* = 15.9 Hz, 1H), 7.07–6.98 (m, 2H), 6.85 (d, *J* = 8.2 Hz, 1H), 6.27 (d, *J* = 15.9 Hz, 1H), 4.26 (s, 4H), and 3.78 (s, 3H); ^13^C NMR (101 MHz, CDCl_3_) δ 167.6, 145.7, 144.4, 143.7, 128.0, 122.0, 117.7, 116.7, 115.8, 64.5, 64.2, and 51.6; ESI-MS *m*/*z*: [M + H]^+^ Calcd for C_12_H_12_O_4_ 221.08 found 221.0; HRMS (EI) *m*/*z*: [M]^+^ Calcd for C_12_H_12_O_4_ 220.0736 found 220.0735.

#### 3.2.2. Synthesis of Methyl (2*S*,3*R*)-3-(2,3-Dihydrobenzo[*b*][1,4]dioxin-6-yl)-2,3-Dihydroxypropanoate (**9**)

To a stirred clear solution of **8** (0.62 g, 2.82 mmol) in *tert*-BuOH (28 mL), water (28 mL), AD-mix-β (3.95 g, 5.07 mmol), and methansulfonaminde (0.29 g, 3.10 mmol) were added. The resulting reaction mixture was stirred for 32 h at room temperature. Sodium sulfite (0.53 g, 4.22 mmol) was added to the reaction mixture and stirred for an additional 2 h at room temperature. The heterogeneous mixture was filtered and rinsed with dichloromethane (DCM). The aqueous layer was extracted with DCM (2 × 50 mL). The combined extracts were dried over anhydrous sodium sulfate, filtered, and concentrated in vacuo. The residue was purified by flash chromatography on silica gel (0~60% EtOAc/*n*-hexanes) to afford **9** (0.59 g) at an 82.4% yield as a white solid (R_f_ = 0.1 (30% EtOAc/*n*-hexanes); mp 96 °C; ${\left[\alpha \right]}_{D}^{25}$ = −33.7 (c 0.1, MeOH); IR(neat) ν 3855, 3809, 3705, 3680, 3411, 2970, 2866, 2843, 2360, 2077, 1734, 1655, 1562, 1456, 1385, 1236, 1055, 1032, 1014, 885, 627, 492, 465, 449, 444, 426, 413, and 403 cm^−1^; ^1^H NMR (400 MHz, DMSO-d_6_) δ 6.84 (s, 1H), 6.77–6.75 (m, 2H), 5.39 (d, *J* = 6.0 Hz, 1H), 5.28 (d, *J* = 7.5 Hz, 1H), 4.71 (dd, *J* = 6.0, 3.9 Hz, 1H), 4.20 (s, 4H), 4.07 (dd, *J* = 7.5, 3.9 Hz, 1H), and 3.58 (s, 3H); ^13^C NMR (101 MHz, DMSO-d_6_) δ 173.2, 143.2, 142.8, 135.6, 119.8, 116.6, 116.0, 76.0, 74.0, 64.5, and 51.9; ESI-MS *m*/*z*: [M + H]^+^ Calcd for C_12_H_14_O_6_ 255.09 found 277; [M + Na]^+^; HRMS (EI) *m*/*z*: [M]^+^ Calcd for C_12_H_14_O_6_ 254.0790 found 254.0786.

#### 3.2.3. Synthesis of Methyl (4*S*,5*R*)-5-(2,3-Dihydrobenzo[*b*][1,4]dioxin-6-yl)-2,2-Dimethyl-1,3-Dioxolane-4-Carboxylate (**10**)

To a solution of **9** (1.46 g, 5.74 mmol) in dry acetone (30 mL), *p*-TsOH (0.1 g, 0.57 mmol) was added. The resulting reaction mixture was stirred for 18 h at room temperature. The mixture was concentrated in vacuo. The residue was purified by flash chromatography on silica gel (0~20% EtOAc/*n*-hexanes) to afford **10** (1.40 g) at an 82.8% yield as a colorless oil (R_f_ = 0.3 (20% EtOAc/*n*-hexanes); ${\left[\alpha \right]}_{D}^{20}$ = +82.6 (c 0.01, MeOH); IR(neat) ν 3855, 3809, 3732, 3410, 2989, 2945, 2831, 2360, 1757, 1593, 1510, 1439, 1381, 1290, 1259, 1209, 1101, 1068, 1026, 889, 818, 492, 464, 457, 444, 432, 424, and 410 cm^−1^; ^1^H NMR (400 MHz, CDCl_3_) δ 6.95 (d, *J* = 1.9 Hz, 1H), 6.92–6.82 (m, 2H), 5.04 (d, *J* = 7.7 Hz, 1H), 4.32 (d, *J* = 7.7 Hz, 1H), 4.26 (d, *J* = 1.2 Hz, 4H), 3.77 (s, 3H), 1.57 (s, 3H), and 1.53 (s, 3H); ^13^C NMR (101 MHz, CDCl_3_) δ 170.7, 143.8, 143.6, 130.7, 119.8, 117.4, 115.5, 111.4, 81.2, 80.4, 64.3, 64.3, 52.4, 26.9, and 25.8; ESI-MS *m*/*z*: [M + H]^+^ Calcd for C_15_H_18_O_6_ 295.12 found 317.0; [M + Na]^+^; HRMS (EI) *m*/*z*: [M]^+^ Calcd for C_15_H_18_O_6_ 294.1103 found 294.1103.

#### 3.2.4. Synthesis of 6-((4*R*,5*R*)-5-(4-Methoxystyryl)-2,2-Dimethyl-1,3-Dioxolan-4-yl)-2,3-Dihydrobenzo[*b*][1,4]di-Oxine (**11**)

To a solution of lithium aluminum hydride (0.89 g, 25.48 mmol) in dry tetrahydrofuran (40 mL), a solution of **10** (2.50 g, 8.49 mmol) in tetrahydrofuran (40 mL) was added at 0 °C. The reaction mixture was stirred for 18 h at 80 °C and quenched very carefully with water at 0 °C. The aqueous layer was extracted with EtOAc (2 × 100 mL). The combined organic layers were dried over anhydrous sodium sulfate, filtered, and concentrated in vacuo to afford crude alcohol. The crude compound was used for the next step without further purification.

To a solution of oxalyl chloride (1.09 mL, 12.74 mmol) in dry dichloromethane (25 mL), dimethyl sulfoxide (1.81 mL, 25.48 mmol) in dichloromethane (25 mL) was added at −78 °C. The reaction mixture was stirred for 1 h at −78 °C. To the reaction mixture, a solution of the alcohol in dichloromethane (25 mL) was added slowly and stirred for another 1 h at the same temperature. Then, trimethylamine (5.92 mL, 42.47 mmol) was added, and the resulting mixture was stirred for an additional 30 min. The reaction mixture was carefully quenched with water. The aqueous layer was extracted with dichloromethane (2 × 100 mL). The combined organic extracts were dried over anhydrous sodium sulfate, filtered, and concentrated in vacuo to afford crude carboxaldehyde.

Freshly prepared 4-methoxybenzyltriphenylphosphonium chloride (3.56 g, 8.49 mmol) was placed in a dry round-bottomed flask under a nitrogen atmosphere and diluted with dry tetrahydrofuran (50 mL). To the reaction mixture, *n*-Buthillithium (4.08 mL, 10.19 mmol, 2.5 M in tetrahydrofuran) was added dropwise at 0 °C. The resulting reaction mixture was stirred for 30 min until the mixture turned to a deep red solution. A solution of aldehyde in dry tetrahydrfuran (25 mL) was added slowly, and the resultant mixture was stirred for 4 h at room temperature. The reaction mixture was carefully quenched with water and extracted with EtOAc (2 × 100 mL). The combined organic layers were dried over anhydrous sodium sulfate, filtered, and concentrated in vacuo. The residue was purified by flash chromatography on silica gel (0~30% EtOAc/*n*-hexanes) to afford **11** (2.1 g, *E*/*Z* = 1:3) at a 67.1% yield (three steps overall) as a colorless oil (R_f_ = 0.33 (20% EtOAc/*n*-hexanes); ${\left[\alpha \right]}_{D}^{20}$ = +92.3 (c 0.01, MeOH); IR(neat) ν 3396, 2981, 2935, 2877, 2835, 1641, 1606, 1508, 1460, 1435, 1373, 1286, 1246, 1165, 1122, 1047, 968, 922, 885, 845, 814, 733, 631, 577, 553, 503, 453, 436, 420, and 407 cm^−1^; ^1^H NMR (400 MHz, CDCl_3_) δ 7.34–7.29 (m, 0.5 H, *E*-form), 6.94 (d, *J* = 1.7 Hz, 1H), 6.89–6.80 (m, 4H), 6.74–6.67 (m, 2H), 6.52 (d, *J* = 15.8 Hz, 0.25 H, *E*-form), 6.06 (dd, *J* = 15.8, 7.4 Hz, 0.25 H, *E*-form), 5.64 (dd, *J* = 11.5, 9.0 Hz, 0.75 H, *Z*-form), 4.66 (dd, *J* = 21.6, 8.4 Hz, 1H), 4.58 (ddd, *J* = 9.0, 8.4, 0.9 Hz, 0.75 H, *Z*-form), 4.32 (ddd, *J* = 8.5, 7.4, 1.0 Hz, 0.25 H, *E*-form), 4.26–4.19 (m, 4H), 3.79 (s, 0.75 H, *E*-form), 3.76 (s, 2.25 H, *Z*-form), 1.58 (d, *J* = 17.0 Hz, 1.50 H, *E*-form), and 1.56 (d, *J* = 3.1 Hz, 4.50 H, *Z*-form); ^13^C NMR (101 MHz, CDCl_3_) δ 159.0, 143.6, 143.6, 135.8, 134.0, 128.4, 127.9, 124.9, 119.8, 117.4, 115.6, 113.4, 109.0, 82.8, 79.1, 64.4, 64.3, 55.2, 27.4, and 27.1 (*Z*-form); 159.5, 143.5, 143.5, 134.0, 130.5, 130.3, 129.0, 122.3, 119.8, 117.2, 115.4, 113.9, 109.0, 84.6, 82.7, 64.4, 64.3, 55.3, 27.3, and 27.2 (*E*-form); ESI-MS *m*/*z*: [M + H]^+^ Calcd for C_22_H_24_O_5_ 369.44 found 391.0; [M + Na]^+^; HRMS (EI) *m*/*z*: [M]^+^ Calcd for C_22_H_24_O_5_ 368.1624 found 368.1620.

#### 3.2.5. Synthesis of (1*R*,2*R*)-1-(2,3-Dihydrobenzo[*b*][1,4]dioxin-6-yl)-4-(4-Methoxyphenyl)but-3-ene-1,2-diol (**12**)

To a solution of **11** (0.7 g, 1.90 mmol) in MeOH (15 mL), 1*N* HCl (5 mL) was added and stirred for 4 h at room temperature. The reaction mixture was concentrated in vacuo. The residue was purified by flash chromatography on silica gel (10~60% EtOAc/*n*-hexanes) to afford **12** (0.50 g) at an 80.1 yield as a yellow oil (R_f_ = 0.17 (30% EtOAc/*n*-hexanes); ${\left[\alpha \right]}_{D}^{20}$ = +121.6 (c 0.01, MeOH); IR(neat) ν 3413, 2933, 2877, 2835, 2249, 1606, 1506, 1460, 1433, 1284, 1246, 1174, 1122, 1065, 1030, 972, 912, 885, 843, 814, 727, 646, 571, 496, 469, 451, 436, 418, and 403 cm^−1^; ^1^H NMR (400 MHz, CDCl_3_) δ 7.07–7.03 (m, 2H), 6.83 (d, *J* = 1.3 Hz, 1H), 6.80–6.77 (m, 4H), 6.51 (d, *J* = 11.7 Hz, 1H), 5.57 (dd, *J* = 11.7, 9.0 Hz, 1H), 4.56–4.51 (m, 1H), 4.49 (d, *J* = 7.0 Hz, 1H), 4.21 (s, 4H), 3.79 (s, 3H), and 3.74–3.72 (m, 2H); ^13^C NMR (101 MHz, CDCl_3_) δ 158.8, 143.3, 143.3, 133.6, 130.0, 130.0, 128.8, 127.7, 119.9, 117.1, 115.9, 113.6, 71.9, 68.0, 64.3, 64.3, and 55.2; ESI-MS *m*/*z*: [M + H]^+^ Calcd for C_19_H_20_O_5_ 329.14 found 351.10; [M + Na]^+^; HRMS (EI) *m*/*z*: [M]^+^ Calcd for C_19_H_20_O_5_ 328.1311 found 328.1309.

#### 3.2.6. Synthesis of 6-((1*R*,2*R*)-1,2-Bis(benzyloxy)-4-(4-methoxyphenyl)but-3-en-1-yl)-2,3-dihydrobenzo[*b*][1,4]di-oxine (**6**)

A solution of **12** (0.40 g, 1.22 mmol) in THF (5 mL) was added to a mixture of sodium hydride (60% dispersed in mineral oil, 0.15 g, 3.78 mmol) and tetra *n*-butylammonium iodide (0.045 g, 0.12 mmol) in THF (10 mL) at room temperature and stirred for 10 min at room temperature. Benzyl bromide (0.43 mL, 3.65 mmol) was added, and the resulting mixture was stirred for 12 h at room temperature. The reaction mixture was quenched with a 50% aqueous NaHCO_3_ solution and extracted with EtOAc (2 × 50 mL). The combined organic layers were dried over anhydrous sodium sulfate, filtered, and concentrated in vacuo. The residue was purified by flash chromatography on silica gel (0~20% EtOAc/*n*-hexanes) to afford **6** (0.44 g) at a 71.0% yield as a yellow oil (R_f_ = 0.28 (20% EtOAc/*n*-hexanes); ${\left[\alpha \right]}_{D}^{20}$ = +57.0 (c 0.01, MeOH); IR(neat) ν 3413, 3060, 3028, 2931, 2870, 2835, 1952, 1873, 1811, 1641, 1606, 1591, 1504, 1454, 1433, 1387, 1348, 1284, 1248, 1203, 1176, 1155, 1092, 1063, 1030, 910, 885, 843, 816, 791, 735, 696, 511, 496, 463, 451, 434, 418, and 407 cm^−1^; ^1^H NMR (400 MHz, CDCl_3_) δ 7.37–7.30 (m, 4H), 7.29–7.25 (m, 1H), 7.21 (dd, *J* = 5.0, 1.9 Hz, 3H), 7.12 (dd, *J* = 6.8, 2.9 Hz, 2H), 7.10–7.04 (m, 2H), 6.90 (d, *J* = 1.8 Hz, 1H), 6.87–6.81 (m, 2H), 6.79–6.74 (m, 2H), 6.62 (d, *J* = 11.7 Hz, 1H), 5.48 (dd, *J* = 11.8, 9.9 Hz, 1H), 4.61 (d, *J* = 12.1 Hz, 1H), 4.58 (dd, *J* = 10.0, 5.6 Hz, 1H), 4.54 (d, *J* = 12.0 Hz, 1H), 4.49 (d, *J* = 5.7 Hz, 1H), 4.37 (d, *J* = 12.1 Hz, 1H), 4.29 (d, *J* = 12.0 Hz, 1H), 4.27 (s, 4H), and 3.80 (s, 3H); ^13^C NMR (101 MHz, CDCl_3_) δ 158.6, 143.2, 143.1, 138.6, 138.5, 133.3, 131.9, 130.0, 129.3, 128.3, 128.0, 127.8, 127.7, 127.7, 127.4, 127.2, 121.1, 116.9, 116.8, 113.4, 83.1, 76.9, 70.8, 70.1, 64.4, 64.3, and 55.3; ESI-MS *m*/*z*: [M + H]^+^ Calcd for C_33_H_32_O_5_ 509.23 found 531.2; [M + Na]^+^; HRMS (EI) *m*/*z*: [M]^+^ Calcd for C_33_H_32_O_5_ 508.2250 found 508.2245.

#### 3.2.7. Synthesis of Benzyl ((1*R*,2*R*)-1-(Benzyloxy)-1-(2,3-dihydrobenzo[*b*][1,4]dioxin-6-yl)-4-(4-Methoxyphenyl)-but-3-en-2-yl)carbamate (**5**)

To a stirred solution of **6** (0.10 g, 0.20 mmol) and sodium carbonate (0.27 g, 2.56 mmol) in a mixture of anhydrous toluene/*n*-hexane (10:1, 4.0 mL), chlorosulfonyl isocyanate (0.34 mL, 3.93 mmol) was added at −78 °C under a N_2_ atmosphere. The reaction mixture was stirred for 24 h at −78 °C and quenched with water very carefully (caution: highly exothermic). The aqueous layer was extracted with EtOAc (2 × 20 mL). The combined organic layers were placed into a 100 mL round-bottomed flask and 25% aqueous solution of sodium sulfite (10 mL) was added. The mixture was further stirred for 12 h at room temperature. The EtOAc layer was separated, and the remaining compound was further extracted with EtOAc (30 mL) from the aqueous layer. The combined organic layers were washed with brine (30 mL), dried over anhydrous sodium sulfate, and concentrated in vacuo. The residue was purified by flash column chromatography on silica gel (0~20% EtOAc/*n*-hexanes) to afford **5** (0.067 g) at a 62.0% yield as a colorless oil (R_f_ = 0.15 (20% EtOAc/*n*-hexanes); ${\left[\alpha \right]}_{D}^{20}$ = +80.8 (c 0.01, MeOH); IR (neat) ν 3413, 3060, 3028, 2931, 2870, 2835, 1952, 1873, 1811, 1641, 1606, 1591, 1504, 1454, 1433, 1387, 1348, 1284, 1248, 1203, 1176, 1155, 1092, 1063, 1030, 910, 885, 843, 816, 791, 735, 696, 511, 496, 463, 451, 434, 418, and 407 cm^−1^; ^1^H NMR (400 MHz, CDCl_3_) δ 7.37–7.33 (m, 3H), 7.31 (dt, *J* = 6.0, 3.4 Hz, 2H), 7.27–7.22 (m, 5H), 7.20 (dd, *J* = 7.7, 1.9 Hz, 3H), 6.77 (d, *J* = 8.2 Hz, 1H), 6.74–6.65 (m, 4H), 6.42 (d, *J* = 11.4 Hz, 1H), 5.74 (dd, *J* = 11.4, 9.8 Hz, 1H), 5.65 (t, *J* = 9.7 Hz, 1H), 5.24 (d, *J* = 12.1 Hz, 1H), 5.14 (d, *J* = 12.1 Hz, 1H), 4.66 (d, *J* = 9.7 Hz, 1H), 4.37 (d, *J* = 11.7 Hz, 1H), 4.25–4.21 (m, 4H), 4.12 (d, *J* = 11.6 Hz, 1H), and 3.77 (s, 3H); ^13^C NMR (101 MHz, CDCl_3_) δ 162.0, 155.7, 143.4, 143.3, 137.9, 136.5, 131.5, 131.3, 128.6, 128.4, 128.4, 128.2, 127.9, 127.9, 127.8, 127.7, 127.1, 126.5, 120.2, 117.1, 116.1, 82.1, 70.9, 66.6, 64.3, 64.3, 62.1, and 58.7; ESI-MS *m*/*z*: [M + H]^+^ Calcd for C_34_H_33_NO_6_ 552.24 found 552.0; HRMS (EI) *m*/*z*: [M]^+^ Calcd for C_34_H_33_NO_6_ 551.2308 found 551.2302.

#### 3.2.8. Synthesis of Benzyl ((1*R*,2*R*)-1-(Benzyloxy)-1-(2,3-Dihydrobenzo[*b*][1,4]dioxin-6-yl)-3-(Pyrrolidin-1-yl)-Propan-2-yl)carbamate (**13**)

To a flame-dried round-bottomed flask containing **5** (0.16 g, 0.30 mmol), dry dichloromethane (15 mL) was added. The solution was cooled to −78 °C, and a stream of O_3_/O_2_ was passed through the reaction for 30 min, after which the reaction was purged with a stream of O_2_ for 1 min. After that, triphenyl phosphine (0.08 g, 0.30 mmol) was added and stirred for 1 h at room temperature. To the reaction mixture, pyrrolidine (0.026 mL, 0.32 mmol) and triethylamine (0.061 mL, 0.44 mmol) were added, and then sodium cyanoborohydride (0.024 g, 0.38 mmol) was added. The reaction mixture was stirred for 16 h at room temperature. The reaction mixture was quenched with water. The aqueous layer was extracted with DCM (2 × 20 mL). The combined organic layers were dried over anhydrous sodium sulfate, filtered, and concentrated. The residue was purified by flash column chromatography on silica gel (0~10% MeOH/EtOAc) to afford **13** (0.12 g) at an 82.3% yield as a colorless oil (R_f_ = 0.43 (10% MeOH/EtOAc); ${\left[\alpha \right]}_{D}^{20}$ = +111 (c 0.01, MeOH); IR (neat) ν 3728, 2942, 2831, 2355, 1658, 1641, 1589, 1512, 1439, 1188, 1119, 1088, 1039, 1026, 721, 692, 525, 476, 457, 449, 438, 422, and 404 cm^−1^; ^1^H NMR (400 MHz, CDCl_3_) δ 7.38–7.24 (m, 10H), 6.86 (q, *J* = 4.0, 3.2 Hz, 1H), 6.85–6.79 (m, 2H), 6.81–6.76 (m, 1H), 5.34 (dd, *J* = 85.0, 9.4 Hz, 1H), 5.14–4.98 (m, 2H), 4.55–4.48 (m, 2H), 4.25 (s, 4H), 4.04 (dt, *J* = 10.0, 5.0 Hz, 1H), 3.10–2.73 (m, 6H), and 1.96–1.85 (m, 4H); ^13^C NMR (101 MHz, CDCl_3_) δ 156.4, 143.7, 143.6, 137.6, 136.4, 128.6, 128.5, 128.1, 128.0, 127.9, 120.0, 117.7, 117.6, 115.8, 115.7, 80.5, 71.2, 70.9, 67.2, 67.1, 64.3, 56.9, 54.5, and 23.3; ESI-MS *m/z*: [M + H]^+^ Calcd for C_30_H_34_N_2_O_5_ 503.26 found 503.2; HRMS (EI) *m*/*z*: [M]^+^ Calcd for C_30_H_34_N_2_O_5_ 502.2468 found 502.2466.

#### 3.2.9. Synthesis of (1*R*,2*R*)-2-Amino-1-(2,3-Dihydrobenzo[*b*][1,4]dioxin-6-yl)-3-(Pyrrolidin-1-yl)propan-1-ol (**14**)

To a solution of **13** (47.0 mg, 0.09 mmol) in ethanol (2.0 mL), 10% palladium on carbon (10 mg) was added, and the reaction mixture was stirred for 16 h under a hydrogen atmosphere. The reaction mixture was filtered through a celite and washed with EtOH (2 × 10 mL). The filtrate was evaporated in vacuo. The residue was purified by reverse phase flash column chromatography with a C18 column (10 ~ 90% MeCN/H_2_O with 0.1% trifluoracetic acid, gradient condition) to afford **14** (21 mg) at an 80.7% yield as a white solid (R_f_ = 0.08 (20% MeOH/EtOAc with 0.05% triethylamine); mp 92.2 °C; ${\left[\alpha \right]}_{D}^{20}$ = +102.3 (c 0.01, MeOH); IR(neat) ν 3855, 3809, 3699, 3680, 3442, 3298, 2970, 2866, 2843, 2827, 2360, 2075, 1736, 1512, 1456, 1211, 1055, 1032, 1014, 646, 492, 465, 455, 442, 430, 413, and 403 cm^−1^; ^1^H NMR (400 MHz, CDCl_3_) δ 6.88–6.75 (m, 3H), 4.55 (s, 1H), 4.25 (s, 4H), 3.15–3.08 (m, 1H), 2.67–2.46 (m, 6H), and 1.76 (s, 4H); ^13^C NMR (101 MHz, CDCl_3_) δ143.3, 142.6, 135.8, 119.2, 116.9, 115.2, 75.8, 64.4, 64.3, 60.1, 54.7, 54.3, and 23.6; ESI-MS *m*/*z*: [M + H]^+^ Calcd for C_15_H_22_N_2_O_3_ 279.17 found 279.1; HRMS (EI) *m*/*z*: [M]^+^ Calcd for C_15_H_22_N_2_O_3_ 278.1630 found 278.1628.

#### 3.2.10. Synthesis of N-((1*R*,2*R*)-1-(2,3-Dihydrobenzo[*b*][1,4]dioxin-6-yl)-1-Hydroxy-3-(Pyrrolidin-1-yl)propan-2-yl)octanamide, Eliglustat (**1**)

To a solution of **14** (17.0 mg, 0.06 mmol) and triethylamine (0.017 mL, 0.12 mmol) in dry DCM (1.2 mL), octanoyl chloride (0.013 mL, 0.08 mmol) was added at 0 °C and stirred for 2 h at the same temperature. The reaction mixture was quenched with water (5 mL), and the aqueous layer was extracted with dichloromethane (2 × 10 mL). The combined organic layers were dried over anhydrous sodium sulfate, filtered, and concentrated in vacuo. The residue was purified by reverse phase flash column chromatography with a C18 column (10~90% MeCN/H_2_O with 0.1% trifluoracetic acid, gradient condition) to afford **1** (13 mg) at a 52.6% yield as a white solid (R_f_ = 0.33 (20% MeOH/EtOAc with 0.05% triethylamine); mp 86.7 °C (lit, mp 87–88 °C) [19]; ${\left[\alpha \right]}_{D}^{20}$ = +124.6 (c 0.01, MeOH); IR(neat) ν 3855, 3809, 3730, 3626, 3369, 2945, 2829, 2360, 1736, 1655, 1512, 1381, 1234, 1053, 1032, 887, 490, 465, 444, 424, 413, and 405 cm^−1^; ^1^H NMR (400 MHz, CDCl_3_) δ 6.84 (d, *J* = 2.0 Hz, 1H), 6.81 (d, *J* = 8.3 Hz, 1H), 6.76 (dd, *J* = 8.3, 2.0 Hz, 1H), 5.81 (d, *J* = 7.5 Hz, 1H), 4.89 (d, *J* = 3.3 Hz, 1H), 4.23 (s, 4H), 4.17 (dddd, *J* = 7.7, 5.6, 4.6, 3.3 Hz, 1H), 2.77 (qd, *J* = 12.9, 5.1 Hz, 2H), 2.69–2.57 (m, 4H), 2.11–2.05 (m, 2H), 1.81–1.72 (m, 4H), 1.50 (p, *J* = 7.5 Hz, 2H), 1.28–1.16 (m, 8H), and 0.85 (t, *J* = 7.1 Hz, 3H) [19]; ^13^C NMR (101 MHz, CDCl_3_) δ 173.4, 143.4 142.8, 134.4, 118.9, 117.0, 115.0, 75.6, 64.3, 57.9, 55.2, 52.2, 36.8, 31.6, 29.1, 29.0, 25.6, 23.6, 22.6, and 14.1; ESI-MS *m*/*z*: [M + H]^+^ Calcd for C_23_H_36_N_2_O_4_ 405.28 found 405.2; HRMS (EI) *m*/*z*: [M]^+^ Calcd for C_23_H_36_N_2_O_4_ 404.2675 found 404.2672.

## 4. Conclusions

In conclusion, we described an asymmetric total synthesis of eliglustat **1** starting from readily available 2,3-dihydrobenzo[*b*][1,4]dioxine-6-carbaldehyde (**7**) via Sharpless asymmetric dihydroxylation and a stereoselective amination reaction using chlorosulfonyl isocyanate (CSI) as key steps. The optimal reaction conditions for the diastereoselective CSI reaction of *syn*-1,2-dibenzyl ether were identified. The retention of the configuration at the *para*-methoxycinnamylic position can be explained by the S_N_i mechanism.

## Data Availability

The data are contained within the article or Appendix A.

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
