# Peer review of "Total Synthesis of Eliglustat via Diastereoselective Amination of Chiral *para*-Methoxycinnamyl Benzyl Ether"

_molecules, 2022, doi:10.3390/molecules27082603_

Round 1
Reviewer 1 Report
The authors present a short and efficient synthesis of Eliglustat, where the key reactions are Sharpless asymmetric dihydroxylation and a chlorosulfonyl isocyanate mediated diastereoselective amination. Overall, the authors have done a great job on the synthesis and presentation of the sequence. This manuscript is suitable for publication in “Molecules” with some minor corrections.
Line 27: please change ‘selective’ to ‘selectively’
Line 51+ Scheme 1 + Scheme 2: Is 7 commercially available? If not, the authors should provide the synthetic steps to 7 or provide citations (if it was reported earlier).
Line 56: Did the authors see only the E-alkene product after HWE reaction?
Line 58: The use of CH3SO2NH2 in case of Sharpless asymmetric dihydroxylation is an unusual condition. The authors should cite the relevant JOC (2009) paper by Junttila et al.
Line 62 + Scheme 2: What is the E:Z ratio for the Wittig olefination step?
Line 71: Please change ‘a toluene solvent’ to ‘toluene’. Did the authors try any polar protic solvent just to see the outcome?
Scheme 3: Please emphasize the dotted bond in the tight ion pair intermediate. It would be visible if the bonds are slightly elongated.
Line 81: It would be better if the ‘four-centered transition state’ is replaced by ‘four-membered transition state’
Author Response
Authors thanks for the reviewer comment.
- Line 27: please change ‘selective’ to ‘selectively’.
Response: According to the reviewer suggestion, ‘selective’ has been changed to ‘selectively’.
- Line 51+ Scheme 1 + Scheme 2: Is 7 commercially available? If not, the authors should provide the synthetic steps to 7 or provide citations (if it was reported earlier).
Response: Compound 7 is commercially available. We have purchased directly from TCI (CAS RN: 29668-44-8).
- Line 56: Did the authors see only the E-alkene product after HWE reaction?
Response: We observed only one product conversion in the TLC. The NMR data support only the E-alkene product.
- Line 58: The use of CH3SO2NH2 in case of Sharpless asymmetric dihydroxylation is an unusual condition. The authors should cite the relevant JOC (2009) paper by Junttila et al.
Response: According to the reviewer suggestion, we have cited the above reference (J. Org. Chem. 2009, 74, 3038-3047) in the manuscript.
- Line 62 + Scheme 2: What is the E:Z ratio for the Wittig olefination step?
Response: The E:Z ratio of the Wittig olefination step is 1:3. We updated the same result in the manuscript.
- Line 71: Please change ‘a toluene solvent’ to ‘toluene’. Did the authors try any polar protic solvent just to see the outcome?
Response: According to the reviewer suggestion, ‘a toluene solvent’ has been changed to ‘toluene’. As we know, CSI consists of two electron-withdrawing components (SO2Cl and -NCO). Due to its resulting electrophilicity, the use of CSI reagent in our synthesis requires relatively inert nonpolar solvents. Therefore, none of any polar protic solvents was tried.
- Scheme 3: Please emphasize the dotted bond in the tight ion pair intermediate. It would be visible if the bonds are slightly elongated.
Response: According to the reviewer suggestion, we revised scheme 3.
- Line 81: It would be better if the ‘four-centered transition state’ is replaced by ‘four-membered transition state’
Response: According to the reviewer suggestion, ‘four-centered transition state’ has been replaced by ‘four-membered transition state’.
Reviewer 2 Report
The manuscript entitled "Total Synthesis of Eliglustat via Diastereoselective Amination of Chiral para-methoxycinnamyl Benzyl Ether"describes the total synthesis of Eliglustat through Sharpless asymmetric dihydroxylation and diastereoselective amination of chiral para-methoxycinnamyl benzyl ethers using chlorosulfonyl isocyanate as the key steps. The research is well-executed and the article is well presented and written. I recommend publication after addressing the below questions:
- The authors have reported a diastereoselectivity level of >20:1 when using toluene/n-hexane (10:1) at -78oC with a yield of 62%. The authors returned this to the increased formation of the tight ion pair intermediate IIA in non-polar solvents (i.e. toluene/n-hexane mixture in this case) rather than the carbocation intermediate IIB. The authors, however, did not explain why reactions carried out in the pure non-polar solvents, toluene and n-hexane (Table 1, entries 4-9), did not return high diastereoselectivity comparable to those obtained when using the mixture of both solvents (Table 1, entries 10-11).
- The authors selected a 10:1 toluene/n-hexane solvent mixture for the reaction, however, no information was provided on the basis upon which this ratio was selected. Further, no information was provided on whether changing this ratio will impact the observed diastereoselectivity of the reaction, to what degree, and why.
- The authors reported a 62% yield for the reaction, however, no information was provided on whether this was due to the reaction not reaching completion or due to the formation of by-products. If it is the first case, did the authors attempt to increase reaction time?! if it is the second case, did the authors attempt to identify the by-products?!
Author Response
- The authors have reported a diastereoselectivity level of >20:1 when using toluene/n-hexane (10:1) at -78oC with a yield of 62%. The authors returned this to the increased formation of the tight ion pair intermediate IIA in non-polar solvents (i.e. toluene/n-hexane mixture in this case) rather than the carbocation intermediate IIB. The authors, however, did not explain why reactions carried out in the pure non-polar solvents, toluene and n-hexane (Table 1, entries 4-9), did not return high diastereoselectivity comparable to those obtained when using the mixture of both solvents (Table 1, entries 10-11).
Response: We could not find an appropriate explanation for the phenomenon of higher diastereoselectivity in the mixed solvent system (toluene/n-hexane) than in other single non-polar solvent systems (toluene or n-hexane only). It was evident that reducing the polarity of the solvent increases the diastereoselectivity when comparing results between dichloromethane and hexane or dichloromethane and toluene at the same reaction temperature. In our previous study, we have tried the mixed toluene/n-hexane solvent system in the CSI amination to prepare syn-configuration (Please refer to I. S. Kim et al., Synlett 2008, 19, 2985-2988). The diastereoselectivity dramatically increased when we adopted the solvent condition (Table 1, entries 10-11). According to our observation, solvent polarity is important for the CSI amination, but unclear factors such as solvent strength or dielectric constant could also determine diastereoselectivity (for example, the dielectric constant order is as follows: DCM (8.93) >> Toluene/n-Hexane (1:10 mixture, 2.56 [calculated by using the method from A. Jouyban et al., J. Chem. Eng. Data 2010, 55, 9, 2951–2963]) > Toluene (2.38) > n-Hexane (1.88)). However, the relationship between the factors and the diastereoselectivity was not examined.
- The authors selected a 10:1 toluene/n-hexane solvent mixture for the reaction, however, no information was provided on the basis upon which this ratio was selected. Further, no information was provided on whether changing this ratio will impact the observed diastereoselectivity of the reaction, to what degree, and why.
Response: The ratio of the 10:1 toluene/n-hexane solvent mixture condition came from our previous study (I. S. Kim et al., Synlett 2008, 19, 2985-2988). We did not observe the impact of changing this ratio. We focused on finding the reaction condition to give better diastereoselectivity rather than discovering the effects of the solvent system.
- The authors reported a 62% yield for the reaction, however, no information was provided on whether this was due to the reaction not reaching completion or due to the formation of by-products. If it is the first case, did the authors attempt to increase reaction time?! if it is the second case, did the authors attempt to identify the by-products?!
Response: The starting material was completely consumed when the reaction was quenched. We also increased reaction time from 24 hours to 72 hours, but more decomposed by-products were observed in longer reaction times. There are decomposed non-polar by-products observed in TLC monitoring. However, separating each by-product from crude is difficult because of multiple spots found in very similar Rf values in TLC.
Reviewer 3 Report
- Authors have nicely designed the total synthesis of Eliglustat using a diastereoslective amination approach using CSI.
- I like the manuscript, though its a very small total synthesis utilizing very well known organic reactions. The major significance of the paper lies in the diastereo selective amination step.
- Authors have mentioned different diastereoselectivity ratios was observed at different reaction conditions, and also mentioned the diastereo selectivity was noted by NMR study. I recommend, authors should provide the NMR spectra with the noted diastereoselectivities of the CSI addition products (at least few) for the readers interest as this is the crucial step for this paper to accept.
- Manuscript was well written and adequate analytical data provided.
- I recommend publication of the article in molecules journal, after authors providing the proper NMR study spectra for the diastereo selectivities of different reaction conditions they used for amination key step.
Author Response
Response: According to the reviewer’s suggestion, 1H NMR spectra for determining diastereoselectivity of CSI amination are provided.
Reviewer 4 Report
Young Hoon Jung group describes the total synthesis of Eliglustat via diastereoselective amination of chiral para-methoxycinnamyl benzyl ether.
This small target molecule has been synthesized very efficiently in many times before (Ref 16-21). The target molecule itself is very straightforward with no complexity involved.
The present study failed to show any improvement in comparison to previous reports. Hence the novelty of this manuscript is poor. As an example; this molecule has been synthesized in six steps with 28.4% overall yield (Ref 20).
The overall yield of the current study is 5% in 12 steps.
This manuscript is not suitable for publication in Molecules.
Author Response
Response: Although other groups have reported the synthesis of Eliglustat, our group mainly focused on synthesizing Eliglustat and chiral syn-amino alcohols via diastereoselective amination of chiral benzylic ethers as a key step using CSI reagent.
Round 2
Reviewer 4 Report
The present study failed to show any improvement in comparison to previous reports. Hence the novelty of this manuscript is poor. As an example; this molecule has been synthesized in six steps with 28.4% overall yield (Ref 20).
The overall yield of the current study is 5% in 12 steps.
This manuscript is not suitable for publication in Molecules.